# An In Situ Experiment to Evaluate the Aging and Degradation Phenomena Induced by Marine Environment Conditions on Commercial Plastic Granules

**DOI:** 10.3390/polym14061111

**Published:** 2022-03-10

**Authors:** Cristina De Monte, Marina Locritani, Silvia Merlino, Lucia Ricci, Agnese Pistolesi, Simona Bronco

**Affiliations:** 1Istituto per i Processi Chimico-Fisici, Sede di Pisa, del Consiglio Nazionale delle Ricerche, (IPCF-CNR), Via G. Moruzzi 1, 56124 Pisa, Italy; cristina.demonte@pi.ipcf.cnr.it (C.D.M.); lucia.ricci@pi.ipcf.cnr.it (L.R.); a.pistolesi@studenti.unipi.it (A.P.); simona.bronco@pi.ipcf.cnr.it (S.B.); 2Istituto Nazionale di Geofisica e Vulcanologia, Via di Vigna Murata 605, 00143 Roma, Italy; 3Istituto di Scienze Marine, Sede di Lerici, del Consiglio Nazionale delle Ricerche, (ISMAR-CNR), 19032 Lerici, Italy; silvia.merlino@sp.ismar.cnr.it; 4Dipartimento di Chimica e Chimica Industriale, Università di Pisa, Via G. Moruzzi 13, 56124 Pisa, Italy

**Keywords:** multi-parametric platform, bioplastics, polymer degradation, marine environment, microplastics, spectroscopy, resin pellets

## Abstract

In this paper, we present two novel experimental setups specifically designed to perform in situ long-term monitoring of the aging behaviour of commercial plastic granules (HDPE, PP, PLA and PBAT). The results of the first six months of a three year monitoring campaign are presented. The two experimental setups consist of: (i) special cages positioned close to the sea floor at a depth of about 10 m, and (ii) a box containing sand exposed to atmospheric agents to simulate the surface of a beach. Starting from March 2020, plastic granules were put into the cages and plunged in seawater and in a sandboxe. Chemical spectroscopic and thermal analyses (GPC, SEM, FTIR-ATR, DSC, TGA) were performed on the granules before and after exposure to natural elements for six months, in order to identify the physical-chemical modifications occurring in marine environmental conditions (both in seawater and in sandy coastal conditions). Changes in colour, surface morphology, chemical composition, thermal properties, molecular weight and polydispersity, showed the different influences of the environmental conditions. Photooxidative reaction pathways were prevalent in the sandbox. Abrasive phenomena acted specially in the sea environment. PLA and PBAT did not show significant degradation after six months, making the possible reduction of marine pollution due to this process negligible.

## 1. Introduction

The United Nations Environment Programme (UNEP) defines “marine litter” as “any persistent, manufactured or processed solid material discarded, disposed of, or abandoned in the marine and coastal environment”, including items made of metals, wood, glass, polymeric materials and some other materials [1]. The amount of marine litter has increased enormously since the middle of the twentieth century, when the awareness of the problem first started to grow, so much so that its size is tentatively considered as one of the factors defining the time boundaries of a new geological era. Many studies confirm that plastic is the main component of marine litter [2,3,4]. However, this quantity can be roughly estimated only and existing reports lead to different results [5].

In 2020, Napper and Thompson established that more than 75% of “marine litters” are plastics [6]. Polymers by their nature do not chemically degrade significantly in the marine environment and, transported by currents, end up accumulating and being deposited on beaches, in the sea (surface, seabed and water column) and even in Arctic Sea ice, depending on the specific density of the materials [7,8,9,10]. For decades, plastic debris in the oceans, coastlines and sea floor has been of great concern as a conspicuous form of pollution [11,12,13,14,15,16,17,18].

In 2021, Ali et al. analysing many studies in recent years, related the abundance of use of various plastics to the abundance of plastic waste dispersed in the environment (air, soil, water) and the consequent negative impact on the environment, animals (in terms of health and food chain), humans (inhalation, ingestion, food chain) and human health [19].

The exposure of macro and meso-plastics (>5 mm) to mechanical, chemical, and biological degradation can lead to their breakdown into smaller pieces at the scale of microplastics, defined as plastic particles with a size of 1 μm to 5 mm [20], or even nanoparticles <1 μm [6,13,21,22,23,24,25,26].

Photooxidation, in particular, leads to the weakening and fracturing of polymers, giving rise to so called “second generation microplastics”. There are also “first generation microplastics” ([5,11,22,27]) that are directly introduced into the marine environment in the form in which they were industrially produced (e.g., microbeads, resin pellets or pre-production granules) [28].

Micro and nanoplastic can be composed of different plastic polymers, such as, among others: polyethylene terephthalate (PET); high-density polyethylene (HDPE); polyvinyl chloride (PVC), which can also vary as rigid PVC, flexible PVC, and polyvinyl acetate (PVAc); low-density polyethylene (LDPE); polypropylene (PP); and polystyrene (PS). Moreover, such plastic is found in a wide variety of structures and shapes, being classified principally as fragments, pellets, filaments, films, foam plastic, granules, and styrofoam [16,29].

The small size of the fragments leads to greater complexity in their recovery and identification, and challenges in the accuracy of the analyses that need to be performed on them [30].

The presence of microplastics in the coastal and marine environment has been widely reported worldwide [18,29], see also [20,31,32,33,34,35,36,37,38,39,40,41,42,43].

In addition to physical damage (e.g., entanglement, suffocation, hypoxia, anoxia [15,39,44,45,46,47,48,49,50,51,52]), several studies have reported toxicological damage caused by the presence of contaminants in the environment (e.g., pesticides, phthalates, PCBs and bisphenol A), adsorbed and released by the plastic debris ingested by the biota and their possible movement along the food chain, with consequences that are still not completely understood [24,53,54,55,56,57,58,59,60,61,62,63,64,65,66,67,68,69,70]. In addition, several plastic items are composed of hazardous monomers and additives (e.g., plasticizer, stabilizer, flame retardants, antioxidants, etc. ([67,71]), and/or they adsorb chemical pollutants from aquatic environments [53,72,73,74,75,76,77].

To date, detrimental effects of microplastics have been reported in fishes [78], mussels [79], meiofauna [80], shrimps [81], crabs [82], and seabirds [83].

In recent years, experimental studies have also focused on the mechanisms of object (especially plastic) degradation [84], and on the connections between degradation and pollutant absorption/desorption mechanisms. The topic is extremely important, because, in addition to its scientific interest, a better understanding of these phenomena could contribute to the development of mitigation actions. It has been stressed by several authors that there are several factors in the sea that lead to a reduction in the speed of degradation. These include lower solar radiation exposure, lower temperatures and changes in salinity [7,22]. Fouling by micro-organisms, limiting the exposure of the surface to UV light, blocks surface photodegradation [85], and reduces the buoyancy of the objects [84], causing a further reduction in exposure to direct sunlight. An understanding of the degradation mechanisms of abandoned plastics in the environment, and in particular in the marine environment, is complicated by variability of the plastics in question, the complexity and extent of the environmental matrix and the boundary conditions. Other issues that cannot be neglected are the mobility and determination of the exact residence times of the materials to be studied.

For example, polyethylene resin pellets kept for eight weeks in the dark, in saltwater under mechanical stirring, showed more morphological and structural changes than under UV exposure [86].

The properties and characteristics of the various polymeric species are related to the mechanisms of chemical and physical degradation (e.g., thermal degradation, photo degradation, hydrolysis) and biological degradation, and the climatic and environmental factors that can affect them, although there is currently no process capable of degrading waste plastic [19].

### 1.1. Bioplastic Case

Several solutions have been proposed in an effort to curb the problem of the increasing presence of plastics in the sea. These include replacing traditional plastics with so-called “bio-plastics” or “compostable plastics”. In part, this has already been implemented in the European Union as a result of EU Directive 2019/904. The goal of the directive was to reduce the consumption of standard single-use plastic items, but, in fact, resulted in the replacement of traditional plastic disposables with biodegradable/compostable plastic disposables. This kind of action does not solve the problem, if it is not accompanied by other equally important actions, such as improving waste management (e.g., sorting, chemical recycling, pyrolysis), preventing waste reaching the sea (e.g., barriers in rivers, improving the transport system and landfills, etc.), and environmental education (e.g., recognizing different types of materials, raising awareness of reuse before recycling). In fact, with implementation of the law, the type of material arriving in the sea changed, with little effect on its quantity. The assumption behind the decision was that biodegradable materials in the sea degrade much faster than traditional materials. However, is this really the case? Some recent studies, focusing on the behaviour of different biodegradable polymers in the environment, cast serious doubt on this.

In 2010, O’Brine and Thompspon [87] compared marine degradation (in Devon, UK) of different plastics, including compostable Mater Bi, two oxo-degradable polyethylenes, and traditional PE, partially from recycled plastics, and observed discharge of 100% of the mass of Mater Bi in 24 weeks, with minimal change observed in the other plastics after 40 weeks. They note that fragmentation of Mater-Bi caused its dispersion in free seawater with no debris being recovered for analysis.

In 2017, Bagheri et al. [88] carried out a study on biodegradable polyesters, including PLGA, PCL, PLA, PHB, Ecoflex and the non-biodegradable polyester polymer PET, maintaining them in artificial seawater and freshwater under controlled conditions for one year. They observed 100% loss only for PLGA, while the other polymers remained intact over 400 days, and concluded that so-called biodegradable polymers do not degrade in water under natural conditions.

In 2020, Kliem et al. [10] analysed the biological degradation of certain polymers under specific conditions of temperature and pH that simulate natural environments, such as soil, fresh water and seawater. They showed that PLA, a biodegradable polymer currently used in industry, does not decompose significantly in seawater or fresh water, but only after industrial composting, with temperature together with battery activity being the most important factors affecting and enabling degradation. Experimental investigations involving biodegradability comparison of chitosan, PBAT (Ecoflex) and HDPE, were carried out in saltwater at the Aquarius underwater laboratory in 2020 [89], with good biodegradation observed for chitosan, 16% of biodegradation for PBAT, and only 7.8 % for HDPE, with FTIR used to evidence structural changes.

Numerous degradation experiments and studies have been performed under controlled conditions and in simulated environments, with only a small number conducted in seawater; however, even in a laboratory environment, it is possible to evaluate photo-degradation and thermal-degradation.

### 1.2. Resin Pellets Case

Among first generation microplastics, in the millimetre range (from 1 to 5 mm) “resin pellets” represent a significant proportion. Pellets are granules of different polymer types, used to produce macroplastic objects, through melting followed by extrusion and moulding. Since there is still no strict regulation for the adoption of measures to prevent the possible loss of these millimetre-sized plastics during their transport, storage and processing, the pellets are easily dispersed in the environment, and are now present in many regions, including the polar regions [90]. Recent studies have shown that their percentage content ranges from 3% [91] to about 30% of all microplastics surveyed on beaches [40,92]. The occurrence of hydrophobic contaminants in marine plastic pellets has been widely reported, and their measurement has established a global association between pollutants and plastic pellets discarded at sea [72,73,74,77,93,94,95,96,97,98,99,100,101].

Laboratory experiments, theoretical modelling and several field experiments have been performed to investigate how the physical and chemical properties of each type of polymer (e.g., surface area, diffusivity, and crystallinity) influence the sorption of chemicals by resin pellets [53,72,95,98,102,103]. The results of these studies confirm that the sorption patterns are dependent on the properties of the particular compound considered (e.g., hydrophobicity, molecular weight (MW)) and on the polymer type, but also on the environmental conditions, such as weathering and water salinity. In many studies, polyethylene (PE) has been shown to present a larger surface area than polypropylene (PP) and to have an affinity for a wide range of organic contaminants varying in hydrophobicity [72,104,105].

PE, together with PP, is produced in very large quantities in west European countries and represents a significant proportion of plastic debris in the environment [75]. This evidence led to the decision to use PE pellets as passive samplers of POP pollution in the International Pellet Watch project [94,97].

Predictive modelling for low-density polyethylene (LDPE) acting as a passive sampling device suggests that the time to reach saturation could vary from days to months, depending on the chemical and environmental conditions considered [97], but also on the sorption/desorption models used, and on the weathering state of the sample [53,95,102].

Most of these studies were carried out in the laboratory, i.e., with the application of precise doses of contaminants in the case of adsorption experiments, or the establishment of controlled conditions of temperature and pressure for degradation experiments. For example, Da Costa et al. [87] studied morphological and thermic variation (using FTIR-ATR analysis, RAMAN spectra, TGA analysis, SEM images) in PE pellets maintained in artificial seawater for eight weeks under controlled temperature, light exposure and movement conditions, and observed quantifiable chemical and physical impacts on the structural and morphological characteristics of granules. They recommended this type of approach to investigate the degradation and behaviour of microplastics.

Such controlled experiments, on the other hand, overlook many factors dependent on environmental and unpredictable weather conditions. Moreover, in laboratory experiments the use of organic-free water prevents a proper assessment of the role of dissolved organic material in the marine environment, for example, by favouring the transfer of chemicals into the pellets, extending the equilibrium time scale for pollutants sorption [103], or influencing weathering mechanisms at the pellet surface.

With respect to the few experiments conducted in free seawater, some of them measured sorption of POPs onto previously uncontaminated plastic pre-production pellets ([73,104]), others focused on degradation of the polymeric matrix [106], while still others focused of the colonization by microorganisms of traditional-plastic and bio-plastic pre-production pellets [107,108,109]. The importance of weathering is emphasized in all these studies. Rochmann et al. [103] focused on the importance of weathering influencing the diffusion behaviour of chemical compounds within the pellets in marine environments. Weathering, leading to an increase of the surface area and pore size of the polymer, promotes an even greater diffusivity in PE pellets, allowing them to sorb a higher quantity of substance. In 2016, Brandon et al. [110] focused only on pellet degradation processes, considering pellets weathered in dry or seawater simulated environment, both in darkness and sunlight conditions. Their results showed non-linear changes in chemical bonds with exposure time to selected environmental conditions.

The long-term effect of atmospheric agents on the mechanisms of degradation of plastics (and, in particular, bioplastics and compostable plastics) in the marine environment, and the connection between degradation and the uptake of chemicals, appears to be a still evolving field of study.

Based on the background described above, our experiment sought to investigate the behaviour of plastic items in the marine environment. These items were both artefacts and pellets of commonly used plastic materials. They were made up of traditional and biodegradable polymers contained in special structures built for the purpose and placed in seawater, or on a simulated beach, and left to age for a minimum of three years. At intervals of a few months, the materials were sampled, and the necessary measurements were taken, to verify any intervening structural and morphological changes. The experiment is still in progress after twenty four months from the positioning of the granules in both environments.

In the Materials and Methods section, we describe the setup of the experiment, the preparation of the cages and samples, the sampling of a portion of the granules of each material after six months, and the types of analyses performed on the samples. The remainder of the granules was left for further sampling in the following months. In the Results section, we report the data obtained for all materials introduced into both the cages and the sandbox, as well as the first results obtained after six months from deployment. In the Discussion section, we comment on these results in the light of what is currently available in the literature. In the Conclusion section, we summarize what has been described above, and describe what our future objectives will be with respect to continuation of the analyses of the material obtained to date, and to continuation of the experiment itself.

## 2. Materials and Methods

### 2.1. Materials

Two types of standard polymer pellets (high density polyethylene—HDPE, Auser Polimeri, Coreglia Antelminelli (LU), Italy, and polypropylene—PP, PoliEko, Celje, Slovenia) were used, together with two types of biodegradable polymer pellets (polylactic acid—PLA, Ingeo 2002D^®^, NatureWorks, Plymouth, MN, USA, and polybutylene adipate-co-terephthalate—PBAT, Ecoflex^®^ F Blend C1200, BASF, Ludwigshafen, Germany). They were used as received.

### 2.2. Experimental Set Up

The experiment was carried out in the Bay of Santa Teresa, a small bay inside the tourist and commercial Gulf of La Spezia, Italy (Figure 1).

The site hosts precious marine and terrestrial ecosystems and lies in very close proximity to the headquarters of the research institutes of the National Research Council (CNR), the National Institute of Geophysics and Volcanology (INGV) and the National Agency for New Technologies, Energy and Sustainable Economic Development (ENEA), which has made the bay a natural laboratory in which to undertake research and evaluate technology. Within this framework, the “Smart Bay Santa Teresa” (https://smartbaysteresa.com/, accessed on 7 February 2022) initiative was inaugurated, a collaboration platform that hosts different scientific projects and an underwater observatory called LabMARE.

This underwater multi-parametric platform was developed by the Ligurian Cluster of the Marine Technologies (DLTM), in collaboration with the INGV, CNR, ENEA, the Hydrographic Institute of the Navy with the support of the Municipality of Lerici and the Cooperative Mitilicoltori Associati, as part of the LabMARE project funded by the Liguria Region.

The main aim of the underwater observatory is experimentation with new technologies and environmental status monitoring. LabMARE is equipped with sensors for monitoring environmental parameters (temperature and salinity) and special cages for studying the degradation of plastics in the marine environment (Figure 2).

The underwater observatory was deployed on 3 March 2020 at 44°4′55.08″ N–9°52′50.46″ E at 10 m depth, about 60 m from the coast (Figure 1) and provides real time temperature data through the underwater cable connected to the land station and via transmission of data by internet connection.

The two special cylindric cages (about 40 cm × 30 cm, Figure 2) installed on the structure of the underwater observatory were self-produced by the researchers in stainless steel 318, each one containing three “baskets” (about 15 cm × 10 cm, Figure 2 in stainless steel wire mesh (AISI 316, mesh 0.24 mm)), electropolished and with anodic passivation. The cages were closed with plastic ties and anchored to the underwater observatory with large plastic ties. Different types of plastic objects, both macro and micro, made of both standard polymers and biopolymers, were placed inside the cages (Table 1). The choice to use metal, avoiding any plastic material to build the cage and baskets was made to prevent any possible contamination, such as possible release of additives from the plastic material (Plastic Additive Standards Guide. AccuStandard, 2018 [106]) or interference with the pellets in the process of absorbing chemicals from the seawater.

Despite the presence of sacrificial anodes placed at one end of each of the two cages, as well as at the end of each of the inner baskets, during August and September 2020, the metal of the cages was affected by corrosion due to a galvanic currents effect. This rapid corrosion was probably due to the large presence of cation metals in the port area. On November 2020, the cages and baskets were changed with new ones constructed by the researchers in collaboration with an enterprise expert in the industrial and naval field (Vamp s.nc., La Spezia): two large cages in stainless steel 316 (size: 70 × 30 cm), 6 small cages in stainless steel 316 (size: 20 × 15 cm) and 8 baskets in stainless steel 316 with meshed net of 1 mm of two different sizes (size: 19 × 14 cm and 14 × 14 cm). Each was equipped with sacrificial anodes to avoid galvanic corrosion. Cages were closed with plastic ties and anchored to the underwater observatory with large plastic ties.

A sandbox containing sand taken from the Gulf of La Spezia (Le Grazie Bay) was set up near the dock where the cages were located, and the same materials introduced into the cages were placed in it, in order to carry out a comparative study of the degradation and absorption of chemicals at sea and on a “simulated beach”. Due to the COVID-19 health emergency, and the consequent total lockdown that prevented free circulation in Italy from February to June 2020, the sandbox was promptly transported to an area near the home (44°7′51.04″ N; 9°57′28.88″ E, in Sarzana) of one of the authors, where it was possible to check that it was not tampered with or damaged.

### 2.3. Experimental Design

The sampling campaign started on 3 March 2020, corresponding to point zero contextually of the first deployment of the LabMARE station and the cages, for all inserted objects, both in the sea cages and in the simulated beach.

The experiment was designed to last three years in total. The objects placed in the cages were of different type, size and material: polymer pellets (see Section 2.1), coffee capsules, single use tableware, personal protective equipment and cigarettes.

In order to study the evolution of ageing of the objects in the different environments, samples were taken on the same day from the cages in the sea and from the sandbox every few months. The sampling dates for the first two years of the experiment were as follows: September 2020, November 2020, March 2021, June 2021, October 2021 and the next sampling is scheduled for the first week of March 2022. All the samples taken in the sea and in the sandbox were washed with ultra-pure water and dried in air before being analysed.

In this investigation, comparison between the characteristics and properties of the raw materials (four types of pellets) and those placed in the two different environments for six months (sampled on September 2020) is discussed. This is the period during which a biodegradable material should be able to biodegrade, according to the UNI EN ISO 14855-2:2018, although the legislation considers composting conditions.

The sampling of material contained in the cages was performed with the support of scubadivers of Dipartimento Polizia di Stat—Centro Nautico Sommozzatori of La Spezia (Italy).

### 2.4. Instruments and Analysis Methods

The marine station LabMARE was equipped with the sensor SBE37SM (supplied by Sea-Bird Scientific) just before the lockdown due to SARS-CoV-2 to record data every 10 min from March 2020. All the data collected were evaluated by calculation of monthly averaged values from the average daily temperatures with relative standard deviations and the maximum and minimum temperatures for each month.

A hygrometer thermometer wireless Bluetooth sensor supplied by ORIA was put in a sandbox located at the CNR research area of Pisa to record the temperature data every 10 min from March 2021. All the data collected were evaluated by calculation of monthly averaged values from the average daily temperatures with relative standard deviations and the maximum and minimum temperatures for each month.

Attenuated total reflectance (ATR) spectra were registered using a Fourier Transform—InfraRed Jasco 6200 (Jasco, Tokyo, Japan) equipped with a PIKE MIRacle (Madison, WT, USA) accessory. Each sample underwent 64 scans from 4000 to 650 cm^−1^, after the collection of background data.

Thermogravimetric analysis (TGA) was performed with an SII TG/DTA 7200 EXSTAR Seiko analyser (Seiko, Chiba, Japa), under heating from 30 to 700 °C, at 10 °C /min rate. Air was fluxed at 200 mL/min during all measurements. The sample amount used for TGA was 5–10 mg. The first derivative (DTG) for each curve was recorded for the different samples and analysed. T_onset_ (temperature corresponding to starting degradation) and residue amount at 700 °C were determined from each TGA profile. T_max_ (temperature at which the maximum rate of mass loss occurs) was calculated from each DTG curve.

Differential scanning calorimetry (DSC) analysis was carried out in order to assess any effects induced by the degradation processes triggered during the months of immersion and in the sandbox relating to plasticisation phenomena or changes in crystallinity. DSC analysis was performed using a Seiko SII ExtarDSC7020 calorimeter (Seiko, Chiba, Japan) with a different thermal programme for each type of polymer (Table 1). Each measurement was carried out on approximately 5–10 mg of materials. The software associated with the instrument was used to determine the values of glass transition temperature (T_g_), thermal capacity variation associated with the glass transition (ΔC_p_), melting temperature (T_m_) with relative enthalpy of fusion (ΔH_m_), and cold crystallization temperature (T_cc_), with relative enthalpy (ΔH_cc_) estimated from the second heating scan. The degree of crystallinity (χ%) was determined using the following Equations (1) and (2) [107]:
(1)$$\chi \%=\frac{\Delta H_{m}}{{\Delta H}_{0m}}\times 100$$
(2)$$\chi \%=\frac{\Delta H_{m}+\;\Delta H_{\mathrm{cc}}}{{\Delta H}_{0m}}\times 100$$
with ΔH_m_ = mean enthalpy of fusion of the sample, ΔH_0m_ = enthalpy of fusion of material 100% crystalline and ΔH_cc_ enthalpy of cold crystallization.

GPC measurements were carried out on PLA and PBAT polymers using an HPLC Agilent 1260 Infinity (Agilent, Santa Clara, CA, United States) equipped with a three-way valve as the injection system and a styrene-divinylbenzene resin as the stationary phase, at an operating pressure of 80 bar, with CHCl_3_ as eluent at 0.3 mL/min and a refractometer as a detector. Solutions were prepared by dissolving the materials in CHCl_3_ (HPLC grade) with a concentration of 3 mg/mL and filtered two times with an Agilent filter with a porosity of 2 μm.

SEM images were recorded with an FEI Quanta 450 ESEM FEG scanning electron microscope (Thermo Fisher, Waltham, MA, USA) at CISUP Laboratories (Centro per la Integrazione della Strumentazione) at University of Pisa. The specimens were analysed on the external surface without any specific manipulations.

## 3. Results

As stated in Section 2, the results obtained from the above-mentioned analyses on standard (HDPE and PP) and biodegradable (PLA and PBAT) polymeric granules during the first six months of the study were reported. The choice of six months was not accidental. The idea was to follow modification due to aging and eventual degradation occurring to the granules in a marine environment (direct immersion in sea or deposition on the surface of the sand in the sandbox) in the first six months after positioning, the period during which a biodegradable material should be able to biodegrade. The results obtained on each kind of “bio” material were compared with the corresponding results for standard polymer granules.

### 3.1. Environmental Analyses

Analysis of the temperatures reached in the sea and in the sandbox in the period under review (March 2020–August 2020) was considered important in order to provide data on whether this variable could affect the characteristics of the aged samples. Table 2 shows the result of the analysis of the data collected at the LabMARE station.

Unfortunately, it was not possible to place a similar sensor in the sandbox near Sarzana (La Spezia). The following were considered in this case. The temperature data were collected in the period from March 2021 to August 2021 using the sensor in the sandbox at CNR and are reported here in Table 3 according to the same analysis carried out for the data in Table 2. In a similar way, the temperatures of the air recorded in Pisa in the period from March 2021 to August 2021 by the San Giusto Weather Station were compared with the data recorded in Sarzana in the period from March 2020 to August 2020 as measured by the Sarzana Luni Weather Station and reported on the website https://www.ilmeteo.it/portale/archivio-meteo, accessed on 7 February 2022 and in Table 4. No significant differences were observed from one year to the other in the air temperatures; the data recorded in the sandbox in Pisa in 2021 were considered consistent with those of the Sarzana sandbox in 2020 in the same months.

Unfortunately, it was not possible to place a similar sensor in the sandbox near Sarzana (La Spezia). The following were considered in this case. The temperature data were collected in the period from March 2021 to August 2021 using the sensor in the sandbox at CNR and are reported here in Table 3 according to the same analysis carried out for the data in Table 2. In a similar way, the temperatures of the air recorded in Pisa in the period from March 2021 to August 2021 by the San Giusto Weather Station were compared with the data recorded in Sarzana in the period from March 2020 to August 2020 as measured by the Sarzana Luni Weather Station and reported on the website https://www.ilmeteo.it/portale/archivio-meteo, accessed on 7 February 2022 and in Table 4. No significant differences were observed from one year to the other in the air temperatures; the data recorded in the sandbox in Pisa in 2021 were considered consistent with those of the Sarzana sandbox in 2020 in the same months.

Given this premise, the analysis of the results indicated that in both the sandbox and the sea similar average temperatures were reached. The substantial difference was the thermal excursion during the day with a maximum of 51 °C in the sandbox and 25 °C in the sea and the corresponding standard deviations. As a consequence, the materials in the sandbox were more thermally stressed than the same materials in the sea.

### 3.2. Six-Months-Aged Standard Polymers

HDPE pellets left in the two different environments (sand and seawater) were analysed using the different techniques described above. The photos collected of the starting granules and of the sampled granules of HDPE after six months in both environments (HDPE_6SW for seawater and HDPE_6S for sand) are compared in Figure 3a.

A colour deviation towards yellow/amber was apparent. For all kinds of granules considered in this experiment, this was much more pronounced in the sea samples. In detail, the effect of an aggressive attack from the environment was evident from the comparison of SEM images (Figure 4a). Modification of the surface morphology taking place correlated with the different action of the sea compared to sand, with formation of an evident porous surface in samples kept in the sea.

These effects were correlated with chemical modification of the polymer chains on the surface of the granules and/or eventual chemical absorption by ATR analyses. New absorption peaks appeared in the region of OH stretching (at about 3400 cm^−1^), between 1740–1550 cm^−1^ due to the stretching mode of carboxyl groups and double bonds in both HDPE_6SW and HDPE_6S (Figure 5a). A large and intense peak between 900–1200 cm^−1^ (centred at about 1010 cm^−1^) was also observed after six months in both environments in the stretching Si-O region.

As far as thermal properties are concerned, TGA and DSC analyses were carried out on the HDPE series. The thermal profiles TGA and DTG are compared in Figure 6a,b and the characteristic temperatures calculated from the thermal curves are summarized in Table 5. A decrease in the thermal stability of the polymer, associated with a decrease in the degradation starting temperature (T_onset_) was observed for both the sea and sand samples after six months. T_max_ decreased only for seawater samples. A slight decrease in the melting temperature for both the seawater and sand samples was observed, as well as in the crystallinity percentage results from the analysis of the second heating scan in the DSC thermogram (Figure 7a and Table 5). The effect was more apparent for HDPE_6SW, as in the TGA results.

The PP granules (Figure 3b) showed a light colour deviation towards yellow which was more pronounced in the seawater sample (PP_6SW) than in the sand sample (PP_6S), as for HDPE. The appearance of some cracks on the surface of the granules was evident in the SEM images of PP_6S (Figure 4b). Formation of absorption bands in the carbonyl and double bond region was observed in the ATR spectra (Figure 5b) and the disappearance of the peak at 1744 cm^−1^ was noticeable. Absorption peaks appeared in the region of OH stretching (centred at about 3420 cm^−1^), both for the sample PP-6SW and PP-6S. As for the HDPE samples, a large and intense peak appearing between 900–1200 cm^−1^ (centred at about 1005 cm^−1^) also formed after six months in both the materials in the stretching Si-O region.

A decrease in T_onset_ and T_max_ with time in both seawater and in sand was observed by comparison of the TGA thermal analyses. An increase in the TGA residue was measured for the PP_6S sample with respect to the other samples (Figure 6c,d and Table 6). A splitting of the T_m_ peak of PP_6S with respect to PP with the presence of a shoulder and a decrease in the crystallinity was measured in the DSC profiles (Figure 7b and Table 6).

### 3.3. Six-Months-Aged Biodegradable Polymers

PLA granules showed a colour deviation towards yellow especially for the sea samples (PLA_6SW) compared to the sand samples (PLA_6S), as previously described for standard polymers (Figure 3c). Noticeable changes in morphology after six months were not observed in both degradative environments (SEM images in Figure 4c). These results were confirmed by ATR analysis. Only a small difference in the spectrum of seawater samples was observed due to the presence of a large peak centred at about 3400 cm^−1^ in the region of -OH stretching (Figure 5c). A broad band between 1700 and 1550 cm^−1^ was detectable in this spectrum with an absorption band also at 1039 cm^−1^.

The results from the ATR and SEM images were confirmed by TGA and DSC analyses (Figure 6e,f and Figure 7c, and Table 7). In the DSC profiles, similarly, the samples showed only minimal variations in the characteristic T_g_, T_m_ and crystallinity degree, statistically not different from those of the PLA.

The molecular weight of the samples and the polydispersity index are given in Table 6. In all cases there was an invariance in polydispersity, even though an improvement in $\bar{Mn}$ and $\bar{Mw}$ values was measured for PLA_6S.

In the case of the PBAT polymer, the effect of the environment on the colour of the pellets was more marked. The pellets developed an amber colour both in the sea (PBAT_6SW) and sand (PBAT_6S) (Figure 3d). The surface of the PBAT granules showed grooves that ran through the granule (Figure 4d). The appearance of a large band in the -OH stretching region (3380 cm^−1^) was evident from ATR analyses (Figure 5d). The presence of additional peaks in the region 1640–1550 cm^−1^ of the double C=C bond was observed in both aged materials, but was more pronounced in the PBAT_6S samples. An intensive peak was also present at 1011 cm^−1^ in the seawater sample.

Regarding the TGA results, the DTG curves in all cases showed two main peaks: the first can be considered the principal degradative step and was used to calculate T_max_, while the second, at higher temperatures, represents a subsequent degradation step which occurred with definitely lower speed. The degradation (Figure 6g,h and Table 8) of the polymer started at a lower temperature for the materials aged in both seawater and in sand. A higher percentage of residue was measured of PBAT_6S. Some effects were also evident in the GPC results, especially for PBAT_6S, where the polydispersity increased from 2.2 of starting material to 2.8 with a decrease in $\bar{Mn}$ and $\bar{Mw}$ (Table 9). The DSC results (Figure 7d and Table 8), however, did not undergo significant change [89].

## 4. Discussion

The polymers chosen for this investigation can be classified, in relation to their chemical structure, into two large families: the C–C backbone and C–O backbone polymers, as reported in Figure 8. HDPE and PP belong to the C–C backbone polymers while PLA and PBAT belong to the C–O backbone polymers.

Overall, this classification translates into a differentiation of properties and chemical behaviour from the point of view of the possibility of being prone to hydrolysis for C–O backbone polymers (PLA and PBAT) or being non-hydrolysable for C–C backbone polymers (HDPE or PP).

HPDE and PP are traditional non-biodegradable plastics and are mainly fossil-fuel based and partly bio-based, as in bio-PE, bio-PP, and bio-PET. However, PLA and PBAT are biodegradable bioplastics, bio-based in the case of PLA (such as PHA or PBS), and fossil-fuel based in the case of PBAT [111].

The effects of the two environments investigated (seawater and sandbox) on the selected materials in the form of granules during the six months are illustrated well by the photos reported in Figure 3. The change in colour can be attributed to several factors, as reported by Andrady et al. [22] and Ali et al. [19], and increased with time of exposure, as observed by Brandon [110]. The colouring (yellowing) of the granules can be mainly related to the start of the degradation process (i.e., photo-oxidation, hydrolysis) or the absorption of pollutants by the surrounding environment [73]. Overall, photo-oxidation and hydrolysis mechanisms lead to the scission of chains with decrease in molecular weight and formation of double bonds C=C and C=O, or the formation of OH groups. Endo et al. [73] hypothesized that the colour change of the granule surface may be due to the facilitated absorption of additives due to the formation of a porous surface with aging. The more pronounced colour deviation towards yellow/amber, found in our study for all sea samples compared to sand samples, seems to indicate that in the first environment the processes causing yellowing are enhanced (whatever they are).

Anyway, noticeable changes in morphology were already evident after six months from the comparison of SEM images (Figure 4) for all the polymers tested in both sea and in sand, with the exception of PLA. The different effects induced by sea or sand conditions on the change in the surface morphology was evident. The formation of a porous surface, or, at least, areas with cavities due to the effect of abrasive phenomena [112], was confirmed in HDPE_6SW and PP_6SW, but particularly in PBAT_6SW. The formation of grooves that ran through the granule observed on the surface of PP_6S granules and PBAT_6S granules can be correlated with the greater exposure of the granule to solar radiation and to the higher temperature reached in the sand environment, which increases with seasonality very differently from sea-water.

The ATR analyses helped to better understand which processes occurred in both granule environments.

The absorption at about 3400 cm^−1^ highlighted in many spectra could be attributed to OH-stretching and was assumed to be the result of a hydrolysis reaction in the polymer chains with an increase in hydroxyl groups.

The peak at about 1010–1030 cm^−1^, shown in the spectra can, alternatively, can be attributed to contamination by sand absorbed on the granule surface (stretching Si-0).

With respect to the standard polymers, the HDPE and PP samples, the appearance of the bands due to the stretching mode of double C=C bond or C=O, that occurred more intensely in PP_6S than HDPE_6S, can be attributed to a more extensive photodegradation process, occurring preferentially in sand compared to seawater. The presence of the methyl sidechain of PP allows β-scission reactions to occur more easily with formation of unsaturated bonds and decrease in molecular weight [113]. The disappearance of the peak at 1744 cm-1 from raw PP can be attributed to additives present in the pristine polymer.

The results appeared to be different for both biodegradable polymers PLA and PBAT. Both PLA_6SW and PLA_6S did not show new peaks, with the exception of that around 3500 cm^−1^. The invariance of the ATR spectra was confirmed by GPC results.

The value of molecular weight remained quite similar, even when the measurements were carried out not by sampling of the surface but on the entire solubilised granule. The percentage of degraded material was low compared to the bulk and did not affect the average molecular weight value [10,88].

Weak peaks appeared in the ATR spectra of the PBAT samples between 1550 and 1640 cm^−1^ which could be associated with C=C stretching, attributable to Norrish II mechanisms [19], and were more intense in the sand environment than in the sea environment.

Materials is not transferred from the surface of the granules to the surface of the crystals after the recording of the ATR spectra and this result indicates the low extent of contamination of the surface of the granules after six months

GPC analysis confirmed these results. An evident decrease was observed in both samples after six months with an increase in polydispersity, especially for the sample in the sandbox.

From the results of the TGA thermal analyses, a decrease in thermal stability was observed for all samples in the sea and in the sandbox, with the greatest degradation occurring for the sandbox samples. These results confirm the higher photodegradation extent occurring for the samples in the sandbox than for those in the sea. The increase in residue percentage for PP_6S (as for PBAT_6S) can be attributed to photo-oxidised samples as reported by Contact-Rodrigo [114].

The DSC results confirmed the findings from the TGA analyses. The splitting of the melting peak in the PP_6S sample and the decrease in the crystallinity can be hypothesized to reflect a possible decrease in the length of the macromolecular chains which could affect the crystallization capacity of the system and therefore give rise to less ordered crystalline forms [114]. On the other hand, Brandon associated increase in brittleness of the material with progression in the environmental exposure time and observed greater action precisely on the PP [110].

The average temperature evolution and temperature values reached during the entire experimental period could explain the limited molecular weight variation for PLA compared to that of PBAT. In the six months of the experiment, the maximum recorded temperature did not exceed 25.24 °C for the sea and 51.63 °C for the sandbox, with a maximum monthly average that reached a maximum only in August 2020 with a value of 23.68 °C (seawater) and 29.06 °C (sandbox). The temperatures recorded remained very far from both the Tg of PLA and Tm temperatures of both polymers.

Unfortunately, while several experiments have been carried out on the biodegradation of PLA (almost always in film form) following the standardized procedures required by international standards (ASTM, UNI EN, ISO) [115,116,117,118], there are few studies regarding the biodegradation of PLA in the marine environment that show little evidence of microbial degradation observed in such environments, temperature conditions and type of microbial species present [119,120].

Although PBAT showed good biodegradation in soil, even at low temperatures [121], PLA showed good biodegradation in compost at 58 °C, at 60% relative humidity and with maintenance of aerobic conditions through oxygen insufflations, while, in contrast, in free soil, its degradation was not significant [118]. Furthermore, in a 100-day experiment to investigate the influence of temperature and type of soil substrates variation on the biodegradation of pure PLA, Gil Castel et al. [116] concluded that chain scission occurring as a function of time and temperature was the main effect influencing the degradation processes of PLA.

## 5. Conclusions

The results of the present study were obtained from the exposure of different types of pre-production pellets to natural sunlight and seawater/beach conditions for six months. At the end of this experiment, we will be able to collect data from a three-year in situ weathering study of standard and bioplastic pellets in the marine environment. Compared to previous studies, based on experiments that accelerated pellet weathering in laboratory or simulated environmental condition, this long-term field experiment represents a more realistic assessment of the aging processes that these particular widespread microplastics experience in the marine environment. As far as we know, this is the first such kind of experiment conducted in the Mediterranean Sea.

In this study, two metal cages containing commercial and commonly used industrial filament plastic granules were placed on a multi-parameter platform at a depth of 10 m in seawater; this was flanked by a sandbox with the same granules placed on the sand surface (simulated beach). The behaviour during aging in the marine environment of both traditional plastics, such as HDPE and PP, and bioplastics, such as PLA and PBAT, was compared.

After six months of experiment (from March to September 2020) the materials immersed in the sea result had been subjected to less thermal stress than the corresponding materials in the sandbox, because of the lower solar radiation and reduced thermal excursion. Prolonged exposure to intense solar radiation and the temperature reached in the sand produced much more visible effects than those produced on samples placed in the depths of the sea in terms of photooxidation by Norrish I and II degradation reactions. On the other hand, the formation of a porous surface and cavities on all six-months-aged seawater samples (except for PLA) was indicative of abrasive phenomena that could induce a greater increase in the absorption of chemicals dissolved in seawater that could, in turn, affect the observed change in colour.

Concerning the biopolymer pellets, the percentage by weight of lost material was low compared to the raw material and did not affect the gravimetric weight value. The molecular weight of PLA remained practically unchanged both in sea and in sand; PBAT, in contrast, showed a decrease of about 40% of the initial molecular weight and less than 10% for the sample in sea. The environmental conditions of the sandbox significantly modified the thermal properties of the materials. A decrease in thermal stability was observed, with the greatest degradation for sandbox samples in all cases and in both environments.

In the two substrates of the experiment (sandbox and sea), the degree of degradation of PLA and PBAT remained low when compared to what was detected for the biodegradation process under composting conditions.

Changes in the chemical-physical properties of the materials suggest material aging and the beginning of a degradation process but with an evolution time that is still long and to be established.

The experiment is still running, and a fuller evaluation of the aging and degradation process will be discussed in the future.

## Figures and Tables

**Figure 1 polymers-14-01111-f001:**
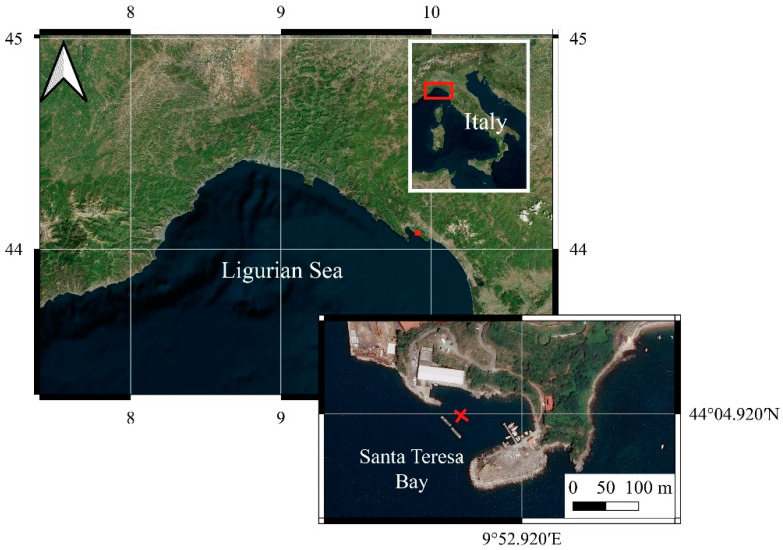
Geographic localization of LabMARE station.

**Figure 2 polymers-14-01111-f002:**
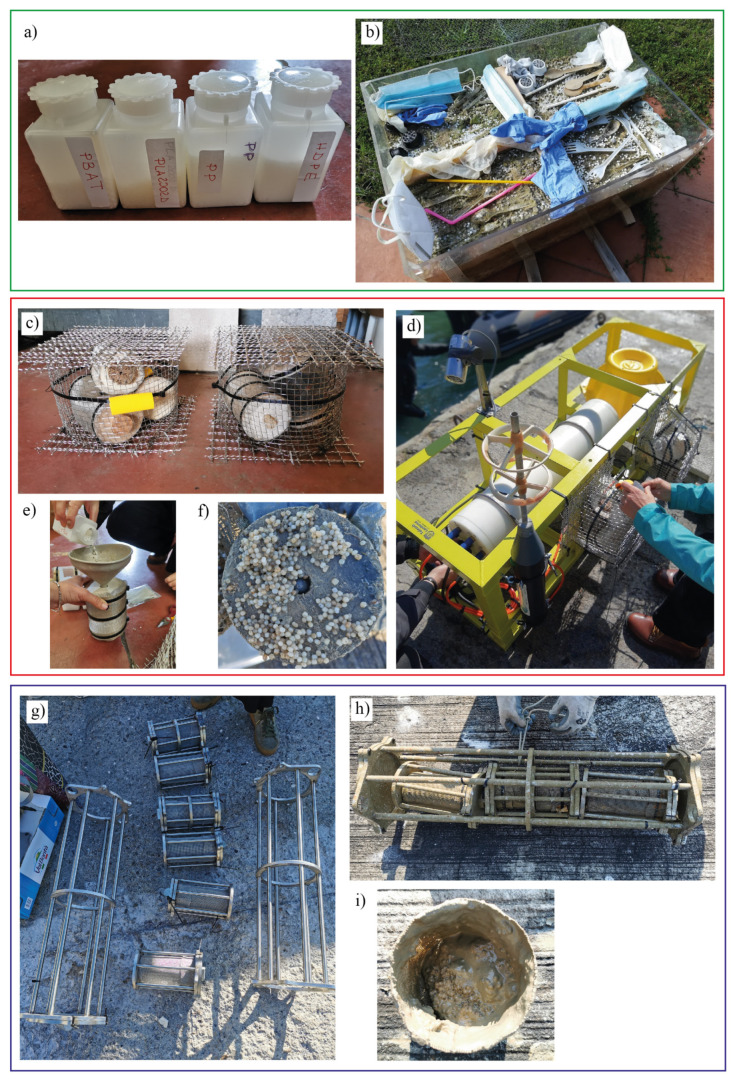
The figure shows: (**a**) containers with raw pellets, (**b**) the sandbox, (**c**) the old, and (**g**) the new version of the cages, (**d**) the underwater observatory before the deployment with the cages installed, (**e**) the insertion of pellets in basket, (**f**,**i**) the pellets in the basket during sampling, (**h**) the content of the cages after first sampling.

**Figure 3 polymers-14-01111-f003:**
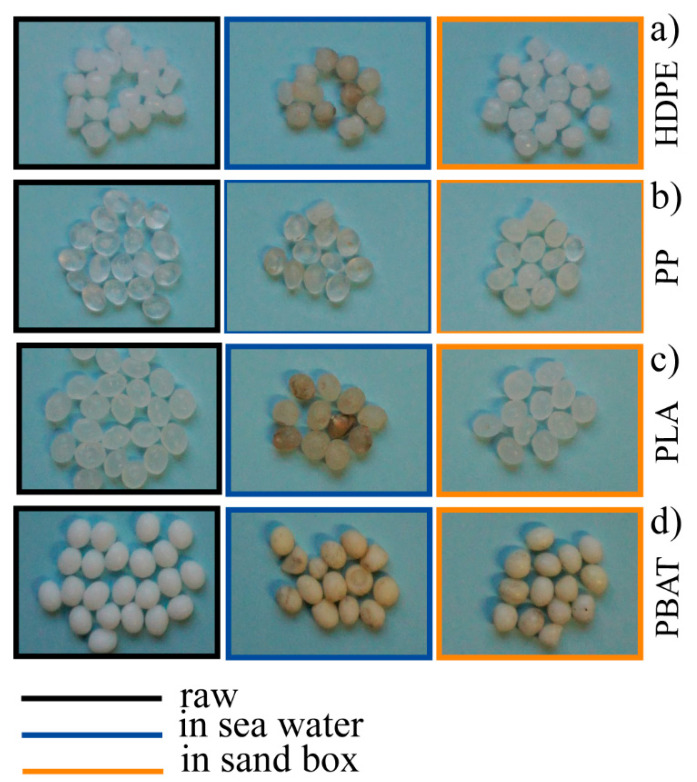
Digital photos of the raw pellets (in black), 6-months-aged in seawater (in blue) and 6-months-aged in sandbox (in orange) (**a**–**d**).

**Figure 4 polymers-14-01111-f004:**
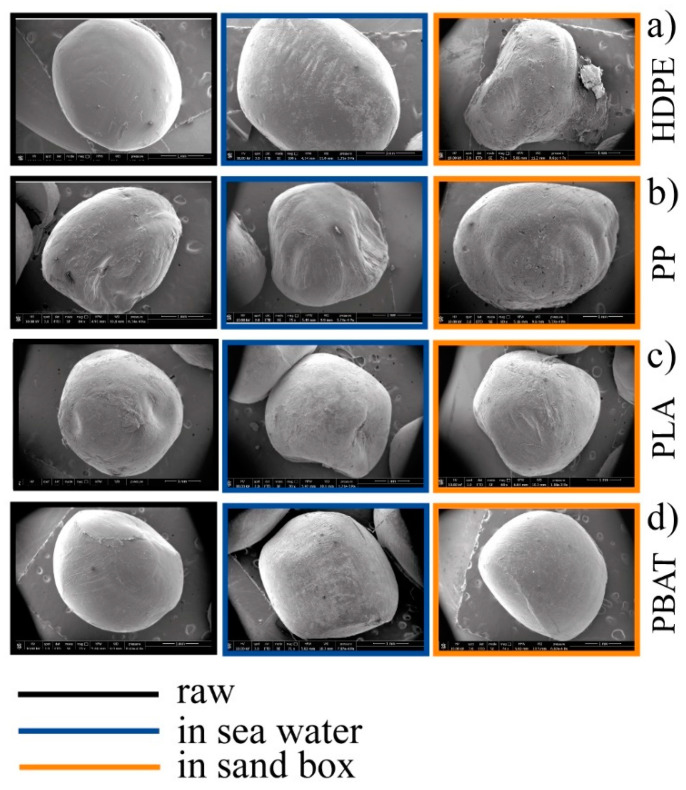
SEM images of the series (**a**–**d**): raw granules in black; 6-months-aged in seawater in blue; 6-months-aged in sandbox in orange.

**Figure 5 polymers-14-01111-f005:**
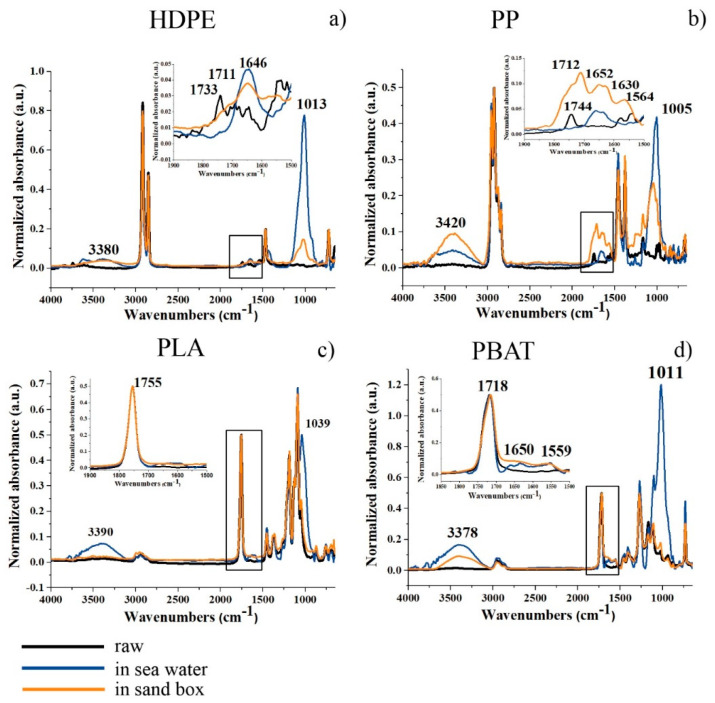
ATR spectra of (**a**–**d**): raw granules in black; six-months-aged in seawater in blue; six-months-aged in sandbox in orange. The inset shows the enlargement of the spectra highlighted from the square.

**Figure 6 polymers-14-01111-f006:**
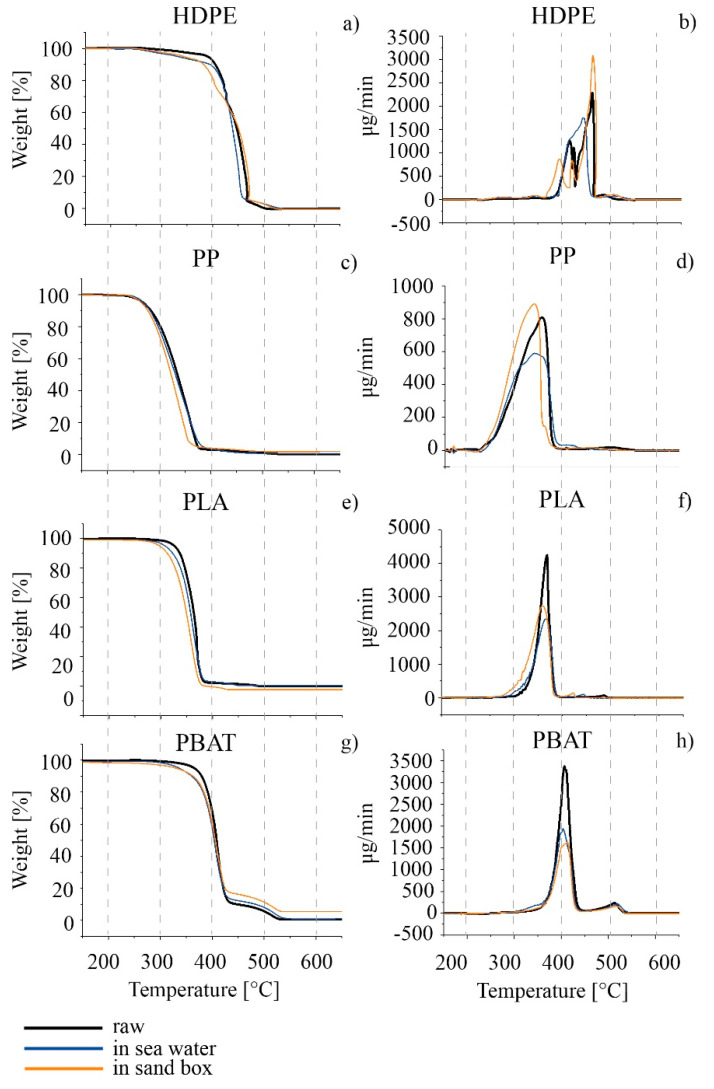
TGA thermograms on the left and the relative DTG on the right of the series (**a**–**h**): raw granules in black; 6-months-aged in seawater in blue; 6-months-aged in sandbox in orange. The scale on the abscissa is the same for all the curves.

**Figure 7 polymers-14-01111-f007:**
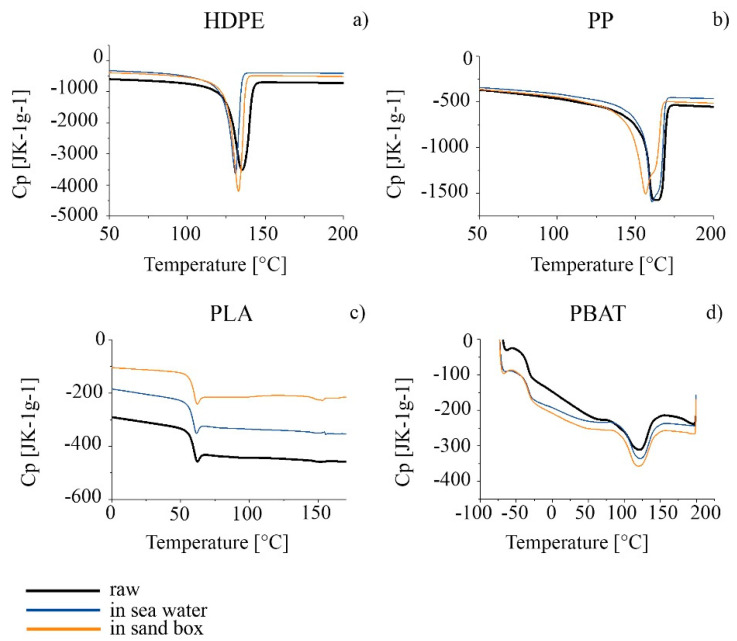
DSC thermograms of the series (**a**–**d**): raw granules in black; 6-months-aged in seawater in blue; 6-months-aged in sandbox in orange.

**Figure 8 polymers-14-01111-f008:**
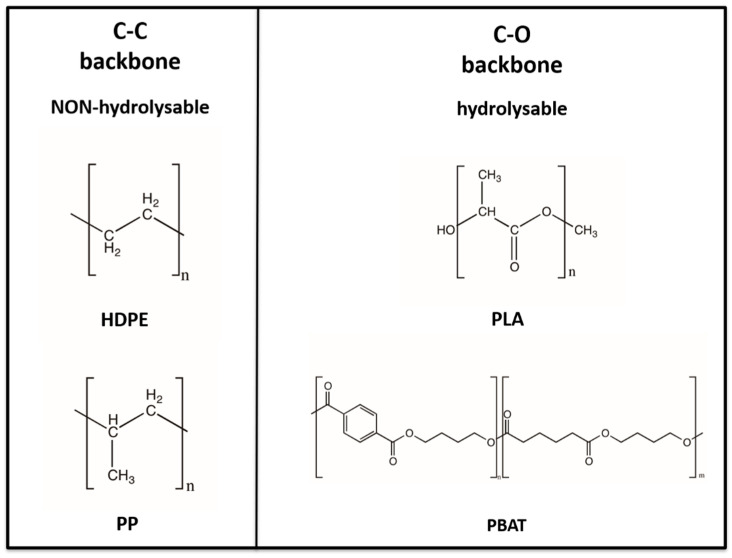
Classification of HDPE, PP, PLA and PBAT based on chemical backbone structure.

**Table 1 polymers-14-01111-t001:** DSC programs for the polymers. Rate of the scans: 10°C/min.

	First Heating (from/to)	Hold (min)	Cooling (from/to)	Hold (min)	Second Heating (from/to)	Hold (min)
HDPE/PP	20−220 °C	2	220/−80 °C	5	−80/220	2
PLA	20−180 °C	1	180/−80 °C	1	−80/180	1
PBAT	20−200 °C	1	200/−80 °C	1	−80/200	1

**Table 2 polymers-14-01111-t002:** Environmental parameters measured in LabMARE.

	Average Temperaures (°C)	Standard Deviation * (°C)	Standard Deviation ** (°C)	Tmin (°C)	Tmax (°C)
March 2020	13.70	0.077	0.32	12.93	14.32
April 2020	15.07	0.055	0.78	13.69	17.33
May 2020	18.20	0.064	0.72	16.93	20.16
June 2020	20.88	0.072	0.85	18.97	23.19
July 2020	21.25	0.11	0.59	19.75	23.35
August 2020	23.68	0.20	2.20	22.12	25.24

* Calculated on daily averages; ** Calculated on all measures.

**Table 3 polymers-14-01111-t003:** Environmental parameters measured in a sandbox placed in Pisa.

	Average Temperaures (°C)	Standard Deviation * (°C)	Standard Deviation ** (°C)	Tmin (°C)	Tmax (°C)
March 2021	12.88	2.94	7.15	1.95	36.16
April 2021	14.98	2.53	6.45	0.123	36.21
May 2021	18.64	2.43	6.05	8.28	42.08
June 2021	25.64	2.15	7.48	11.79	45.60
July 2021	27.69	1.99	6.82	17.84	48.69
August 2021	29.06	2.09	7.19	18.81	51.63

* Calculated on daily averages; ** Calculated on all measures.

**Table 4 polymers-14-01111-t004:** Comparison of the air temperature data of Pisa and Sarzana.

Sarzana	Average Temperaures (°C)	Standard Deviation (°C)	T_min_ (°C)	T_max_ (°C)	Pisa	Average Temperaures (°C)	Standard Deviation (°C)	T_min_ (°C)	T_max_ (°C)
March 2020	11.09	2.22	2	20	March 2021	9.65	1.85	−1	22
April 2020	14.48	2.58	2	21	April 2021	11.70	2.15	−2	21
May 2020	19.35	1.98	12	25	May 2021	15.39	1.67	5	24
June 2020	20.67	2.67	14	30	June 2021	21.93	1.86	10	32
July 2020	24.65	1.53	18	30	July 2021	24.23	1.89	16	33
August 2020	25.19	1.86	18	32	August 2021	24.74	1.97	15	36

**Table 5 polymers-14-01111-t005:** Significant temperatures from TGA and DSC curves of HDPE samples.

	TGA Results			DSC Results		
Sample	T_onset_ (°C)	T_max_ (°C)	Residue at 700 °C (%)	T_m_ (°C)	ΔH_m_ (J/g)	χ% *
HDPE	255.5	464.3	0.8	135.1	228	77.8
HDPE_6SW	249.1	444.7	0.0	130.9	183	62.5
HDPE_6S	242.1	465.0	0.0	132.8	207	70.6

* ΔH^0^_m_ = 293 J/g [108]; $\chi \%=\frac{\Delta H_{m}}{{\Delta H}_{0m}}\times 100$.

**Table 6 polymers-14-01111-t006:** TGA and DSC results of PP samples.

	TGA Results			DSC Results		
Sample	T_onset_ (°C)	T_max_ (°C)	Residue at 700 °C (%)	T_m_ (°C)	ΔH_m_ (J/g)	χ% *
PP	261.5	355.7	0.3	162.5	108.0	52.2
PP_6SW	261.4	340.6	0.9	160.6	102.0	49.3
PP_6S	255.0	339.4	2.2	156.3 (162.4)	98.6	47.6

* ΔH^0^_m_ = 207 J/g [108]; $\chi \%=\frac{\Delta H_{m}}{{\Delta H}_{0m}}\times 100$.

**Table 7 polymers-14-01111-t007:** TGA and DSC results of PLA samples.

	TGA Results			DSC Results						
Sample	T_onset_ (°C)	T_max_ (°C)	Residue at 700 °C (%)	T_g_ (°C)	ΔC_p_ (J/g°C)	Tcc (°C)	ΔHcc (J/g)	T_m_ (°C)	ΔH_m_ (J/g)	χ% *
PLA	323.1	367.7	0.0	58.3	0.55	122.7	−0.19	150.9	0.21	0.022
PLA_6SW	320.2	365.4	0.6	57.5	0.57	-	-	147.9	0.16	0.17
PLA_6S	315.0	359.2	0.0	58.5	0.51	117.6	−1.89	147.5 (152.4)	2.04	0.16

* ΔH^0^_m_ = 93 J/g [109]; $\chi \%=\frac{\Delta H_{m}+\;\Delta H_{\mathrm{cc}}}{{\Delta H}_{0m}}\times 100$.

**Table 8 polymers-14-01111-t008:** TGA and DSC results of PBAT samples.

	TGA Results			DSC Results				
Sample	T_onset_ (°C)	T_max_ (°C)	Residue at 700 °C (%)	T_g_ (°C)	ΔC_p_ (J/g°C)	T_m_ (°C)	ΔH_m_ (J/g)	χ% *
PBAT	369.9	407.1	0.5	−35.4	0.40	120.6	18.4	16.1
PBAT_6SW	359.3	403.6	0.6	−35.0	0.41	122.1	18.2	16.0
PBAT_6S	357.2	412.6	5.6	−36.0	0.44	119.5	18.7	16.4

* ΔH^0^_m_= 114 J/g [109];$\;\chi \%=\frac{\Delta H_{m}}{{\Delta H}_{0m}}\times 100$.

**Table 9 polymers-14-01111-t009:** GPC results of PLA and PBAT samples.

Sample	$\bar{Mn}$ (KDa)	$\bar{Mw}$ (KDa)	PDI
PLA	84.6	146.3	1.7
PLA_6SW	82.2	145.2	1.8
PLA_6S	88.9	149.7	1.7
PBAT	21.5	47.3	2.2
PBAT_6SW	19.8	45.2	2.3
PBAT_6S	12.6	35.8	2.8

## Data Availability

The data that support the findings of this study are available from the authors upon reasonable request.

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
