# Peer review of "An In Situ Experiment to Evaluate the Aging and Degradation Phenomena Induced by Marine Environment Conditions on Commercial Plastic Granules"

_polymers, 2022, doi:10.3390/polym14061111_

Round 1
Reviewer 1 Report
Dear authors
The subject of the article is interesting and up-to-date. It concerns the degradation of polymeric materials in seawater. It is directly related to the protection of the environment and indirectly with human health.
In the introductory part, the Authors presented the main mechanisms of the destruction of polymeric materials, mainly as a result of the phenomena of photooxidation and adsorption, desorption of various types of pollutants, and characterized the influence of temperature. As a result of their destruction, these materials in the form of microparticles contribute to the deterioration of the health of living organisms. In the introduction, attention was drawn to the actions taken by international institutions to reduce the amount of microplastics in seas and oceans. In the article, the authors undertook the quantitative and qualitative assessment of the impact of sea water on the aging processes of plastics such as HDPE, PP, PLA, PBAT. It was found that sea water does not significantly accelerate the degradation process of these The research methodology was developed correctly, adequately to the formulated goal of the research work. materials. The test results were processed correctly. The conclusions drawn from the research are justified in the content of the article.
I have three minor comments regarding the names used:
Page 7
There is “ passivated by electropolishing”.
In my opinion, it should be: electropolishing and anodic passivation
or electrochemical passivation”
Page 8
There is sentence: “This rapid corrosion has been probably due to the large presence of metal in the port area”
In my opinion, it should be: This rapid corrosion has been probably due to the large presence of cations metals in the port area.
Page 9
There is sentence: “SEM images were recorded with a FEI Quanta 450 ESEM FEG optical scanning microscope at CISUP Laboratories at University of Pisa".
In my opinion, it should be: SEM images were recorded with a FEI Quanta 450 ESEM FEG scanning electron microscope at CISUP Laboratories at University of Pisa.
Sincerely
Author Response
Response to Reviewer 1 Comments
Ref: POLYMERS-1611545
Title (the old one): The aging and degradation phenomena induced by marine environment conditions on commercial plastics granules
Journal: Polymers-Special Issue: Microplastics Degradation and Characterization
Dear Reviewer,
please find enclosed our comments and revision of the manuscript previously titled “The aging and degradation phenomena induced by marine environment conditions on commercial plastics granules” by Cristina De Monte, Marina Locritani, Silvia Merlino, Lucia Ricci, Agnese Pistolesi and Simona Bronco according to the indications received from the referees.
Changes have been made to the submitted document and include suggestions from both reviewers. In details, our revision and comments are reported below (in italics and in red) after the comments of the editor and referees (in black). Regarding the specific issues addressed the authors respond below specifying in which line the changes were made.
Specific comments:
The subject of the article is interesting and up-to-date. It concerns the degradation of polymeric materials in seawater. It is directly related to the protection of the environment and indirectly with human health.
In the introductory part, the Authors presented the main mechanisms of the destruction of polymeric materials, mainly as a result of the phenomena of photooxidation and adsorption, desorption of various types of pollutants, and characterized the influence of temperature. As a result of their destruction, these materials in the form of microparticles contribute to the deterioration of the health of living organisms. In the introduction, attention was drawn to the actions taken by international institutions to reduce the amount of microplastics in seas and oceans. In the article, the authors undertook the quantitative and qualitative assessment of the impact of sea water on the aging processes of plastics such as HDPE, PP, PLA, PBAT. It was found that sea water does not significantly accelerate the degradation process of these. The research methodology was developed correctly, adequately to the formulated goal of the research work. materials. The test results were processed correctly. The conclusions drawn from the research are justified in the content of the article.
The authors thank the reviewer for valuable advice and appreciation of the topic covered in the paper.
Changes have been made to the submitted document and include suggestions from both reviewers.
Page 1 line 4-6 old document; Page 1 line 4-6 new document: Title changed in: An in situ experiment to evaluate the aging and degradation phenomena induced by marine environment conditions on commercial plastic granules.
Page 7 line 304 old document: There is “passivated by electropolishing”. In my opinion, it should be: electropolishing and anodic passivation or electrochemical passivation”
Page 7 line 315 new document : As suggested by reviewer the author change “passivated by electropolishing” in “electropolishing and anodic passivation”.
Page 8 line 316 old document: There is sentence: “This rapid corrosion has been probably due to the large presence of metal in the port area”. In my opinion, it should be: This rapid corrosion has been probably due to the large presence of cations metals in the port area.
Page 8 Line 328 new document: As suggested by reviewer the author add “cations”.
Page 9 line 400 old document: There is sentence: “SEM images were recorded with a FEI Quanta 450 ESEM FEG optical scanning microscope at CISUP Laboratories at University of Pisa". In my opinion, it should be: SEM images were recorded with a FEI Quanta 450 ESEM FEG scanning electron microscope at CISUP Laboratories at University of Pisa.
Page 10 Line 426 new document: As suggested by reviewer the author add “electron” and removed “optical”.
Best regards,
Marina Locritani

Reviewer 2 Report
This paper is worthy of inclusion in the proposed special issue after major revision and careful attention to the English language in many parts of the paper. For example the Abstract is very poor but editing by a native English speaker would soon address this. There are also inconsistencies eg an abbreviation "PGDA" is used on page 3 but this does not correspond to any polymer discussed.
Marine pollution by polymer granules is a challenging area because of the physical size of the material and the fact that these particles may contain quite high levels of additives which greatly extend the lifetime in the environment. This paper is a study after only 6 months of exposure and would be of more interest if continued for several years. As such this must be regarded as a preliminary study but is interesting as it shows the challenges of doing marine exposure.
As a study in degradation of polymers the paper would be rejected as it lacks scientific depth and the starting materials need much greater characterization regarding the additives which will dominate the performance in the first 6 months. The section on FTIR is poor. The results are discussed at a superficial level and need some expert input. In ATR, surface contamination often dominates the spectrum of "real world" samples when removed from the field (a simple water wash is not enough) and spectra should be collected again after the sample is removed from the ATR element to determine what has been transferred under the pressure of contact. The spectra presented require great expansion to see the detail but when so examined they are very typical of such surface contamination and it is very misleading to suggest they indicate oxidation. This is most unlikely. The strong band at 1000 cm-1 is a clear indication of contamination (it is often Si-O in many different forms) and during polymer oxidation such a band does not form due to carbonyl, carboxylate and double bond formation as the authors suggest.
The authors should also at some point discuss why PLA and PBAT will not show degradation under these conditions (marine or sandbox) while they will show considerable degradation in 6 months under composting conditions. This requires the degradation temperature to approach the glass transition temperature of the polymer so that hydrolytic chain scission under active microbial exposure can occur. This is unlikely to occur in the marine or sandbox environments. This could be a useful point to be made in the paper regarding the different meaning of "biodegradable" polymers in different environments.
Overall the scientific shortcomings are outweighed by the interest in the topic so I recommend publication after Major Revision.
Author Response
Response to Reviewer 2 Comments
Ref: POLYMERS-1611545
Title (the old one): The aging and degradation phenomena induced by marine environment conditions on commercial plastics granules
Journal: Polymers-Special Issue: Microplastics Degradation and Characterization
Dear Reviewer,
please find enclosed our comments and revision of the manuscript previously titled “The aging and degradation phenomena induced by marine environment conditions on commercial plastics granules” by Cristina De Monte, Marina Locritani, Silvia Merlino, Lucia Ricci, Agnese Pistolesi and Simona Bronco according to the indications received from the referees.
Changes have been made to the submitted document and include suggestions from both reviewers. In details, our revision and comments are reported below (in italics and in red) after the comments of the editor and referees (in black). Regarding the specific issues addressed the authors respond below specifying in which line the changes were made.
Specific comments:
This paper is worthy of inclusion in the proposed special issue after major revision and careful attention to the English language in many parts of the paper. For example the Abstract is very poor but editing by a native English speaker would soon address this.
The title have been modified and the English of the abstract and the entire text checked by an expert. Regarding the specific issues addressed the authors respond below specifying in which line the changes were made. Changes have been made to the submitted document and include suggestions from both reviewers.
Page 1 line 4-6 old document; Page 1 line 4-6 new document: Title changed in: An in situ experiment to evaluate the aging and degradation phenomena induced by marine environment conditions on commercial plastic granules.
There are also inconsistencies eg an abbreviation "PGDA" is used on page 3 but this does not correspond to any polymer discussed.
Page 3 Line 147 old document: The author insert the acronym “PLGA” and removed poly(lactic-co-glycolic acid)
Page 3 line 148 and 151 new document: The author revised the typing-mistake of “PGDA” with “PLGA”
Marine pollution by polymer granules is a challenging area because of the physical size of the material and the fact that these particles may contain quite high levels of additives which greatly extend the lifetime in the environment. This paper is a study after only 6 months of exposure and would be of more interest if continued for several years. As such this must be regarded as a preliminary study but is interesting as it shows the challenges of doing marine exposure.
The authors have better specified more fully the design of the experiment which is a long-term experiment in deep-sea and sandbox in section 1.2 and in section 2.3 and that the results described in the submitted manuscript were collected in the first 6 months of the experiment which is still in progress and will conclude after three years in total.
The author outline here that the following results on the development of the aging and degradation phenomena observed after the first 6 months here reported will be the object of further manuscript. In details:
Page 5 Line 245 new document: The authors added the sentence “and left to age for minimum three years”
Page 5 Line 247-249: The authors added the sentence “Actually, the experiment is still in progress after 24 months from the positioning of the granules in both environments”.
Page 5 Line 252-255: The author added the sentence “the sampling of a portion of the granules of each materials after six months” and “The rest of the granules was left further samples in the following months”
As a study in degradation of polymers the paper would be rejected as it lacks scientific depth and the starting materials need much greater characterization regarding the additives which will dominate the performance in the first 6 months.
The authors agree with the reviewer that at this stage of the long-term experiment, the results outline largely aging effects and only initial degradation phenomena. The evidences of the degradation process could be much clearer and better described in the following studies still in progress in both environments.
Page 8 Line 351-367: The authors elaborated a more detailed information on the experimental design as requested by the reviewer.
Page 10 Line 405 old document – Page 10 line 430 new document: The authors substitute “In spite of the experimental design “ with “As already said in Section 2”
Page 10 Line 409: The authors delete “and or degradation”
Page 10 Line 412-414 old document: The authors move the sentence “according to the UNI EN ISO 14855-2:2018, even if the legislations consider composting conditions”to Page 8 line 366 new document.
The section on FTIR is poor. The results are discussed at a superficial level and need some expert input. In ATR, surface contamination often dominates the spectrum of "real world" samples when removed from the field (a simple water wash is not enough) and spectra should be collected again after the sample is removed from the ATR element to determine what has been transferred under the pressure of contact. The spectra presented require great expansion to see the detail but when so examined they are very typical of such surface contamination and it is very misleading to suggest they indicate oxidation. This is most unlikely. The strong band at 1000 cm-1 is a clear indication of contamination (it is often Si-O in many different forms) and during polymer oxidation such a band does not form due to carbonyl, carboxylate and double bond formation as the authors suggest.
The authors want to justify with the reviewer because they forgot the sentence now reported in the revision about the presence of silica on the surface of the granules after simple washing with water. This carelessness has as a consequence a wrong explanation of the spectra. The authors want to thank the reviewer for the comment.
In section 3.2 (HDPE and PP) and 3.3 (PLA and PBAT) the much more significant absorption bands of the ATR spectra are reported and then commented in details in section 4. In details the new paragraphs are reported at:
Page 13 Line 475-479 old document – Page 13 line 498-504 new document: the description of ATR spectra of HDPE is rewritten
Page 16 Line 507-511 old document – page 16 line 535-542 new document: the description of ATR spectra of PP is rewritten
Page 17 Line 530-532 old document – Page 17 line 561-565 new document: the description of ATR spectra of PLA is rewritten
Page 17 Line 543-547 old document – Page 17 line 576-581 new document: the description of ATR spectra of PBAT is rewritten
Page 20 Line 610-662 old document – page 20 line 644-673 new document: the discussion of ATR spectra of HDPE, PP, PLA and PBAT is rewritten
Page 20 line 691-714 new document: All the final part of the discussion (Section 4) have been revised
Anyway, the authors would like to outline that no transfer of significant amount of material from the granules to the surface of the ATR crystals occurs during the measurements as it is demonstrated by the picture here below as an example (spectrum of HDPE_6SW). In red is reported the spectrum of the surface of crystals after the removal of the granules and in black the spectrum of the granules. If it is requested, the authors can give also the other comparisons of all samples.
Page 14 line505-506 new document: the author modified Fig. 5 ATR spectra.
The authors should also at some point discuss why PLA and PBAT will not show degradation under these conditions (marine or sandbox) while they will show considerable degradation in 6 months under composting conditions. This requires the degradation temperature to approach the glass transition temperature of the polymer so that hydrolytic chain scission under active microbial exposure can occur. This is unlikely to occur in the marine or sandbox environments. This could be a useful point to be made in the paper regarding the different meaning of "biodegradable" polymers in different environments.
Page 20 Line 699-714 new document: The authors agree with the referee and add a paragraph discussing the different response of biodegradable materials to different environmental conditions.
Page 21 line 716-726 new document: the authors added short paragraph in the Conclusion section.
Overall the scientific shortcomings are outweighed by the interest in the topic so I recommend publication after Major Revision.
Response:
The authors thank the reviewer for valuable advice and appreciation of the topic covered in the paper.
Best regards,
Marina Locritani

Round 2
Reviewer 2 Report
The authors have thoroughly addressed all of the issues that I raised.
I am happy to recommend acceptance.